# COPA-SSE: Semi-Structured Explanations
# for Commonsense Reasoning

**Ana Brassard**[1, 2]                    ANA.BRASSARD@RIKEN.JP

**Benjamin Heinzerling**[1, 2]            BENJAMIN.HEINZERLING@RIKEN.JP

**Pride Kavumba**[2, 1]                    PKAVUMBA@ECEI.TOHOKU.AC.JP

**Kentaro Inui**[2, 1]                     INUI@ECEI.TOHOKU.AC.JP

[1]*Riken AIP*

[2]*Tohoku NLP Lab, Tohoku University*

## Abstract

We present Semi-Structured Explanations for COPA (COPA-SSE), a new crowdsourced dataset of 9,747 semi-structured, English common sense explanations for COPA questions. The explanations are formatted as a set of triple-like common sense statements with ConceptNet relations but freely written concepts. This semi-strucutred format strikes a balance between the high quality but low coverage of structured data and the lower quality but high coverage of free-form crowdsourcing. Each explanation also includes a set of human-given quality ratings. With their familiar format, the explanations are geared towards commonsense reasoners operating on knowledge graphs and serve as a starting point for ongoing work on improving such systems.

## 1. Introduction

While there are many datasets for question answering and commonsense reasoning [Rogers et al., 2021], models are known to exploit shortcuts such as superficial cues in these datasets, which leads to artificially high evaluation scores [Gururangan et al., 2018]. One way to ensure models are reasoning as intended is to require explanations for their predictions [Bowman and Dahl, 2021]. A prominent example of such a setting is the Commonsense Explanations Dataset (CoS-E) [Rajani et al., 2019], which provides crowdsourced justifications of the correct answers expressed in free text. While free-form crowdsourcing allows representing natural and diverse human reasoning, quality control is notoriously difficult [Daniel et al., 2018]. At the other end of the spectrum are explanations that are fully grounded in a knowledge graph (KG), i.e., each element of the explanation corresponds to a node or edge in a KG. However, this structured approach is limited by the coverage of the KG, i.e., the explanation will be suboptimal or impossible when the situation to explain is not covered by the KG. Here, we adopt a semi-structured approach aiming to combine the best of both worlds—the coverage potential of open-ended crowdsourcing and quality control of structured data.

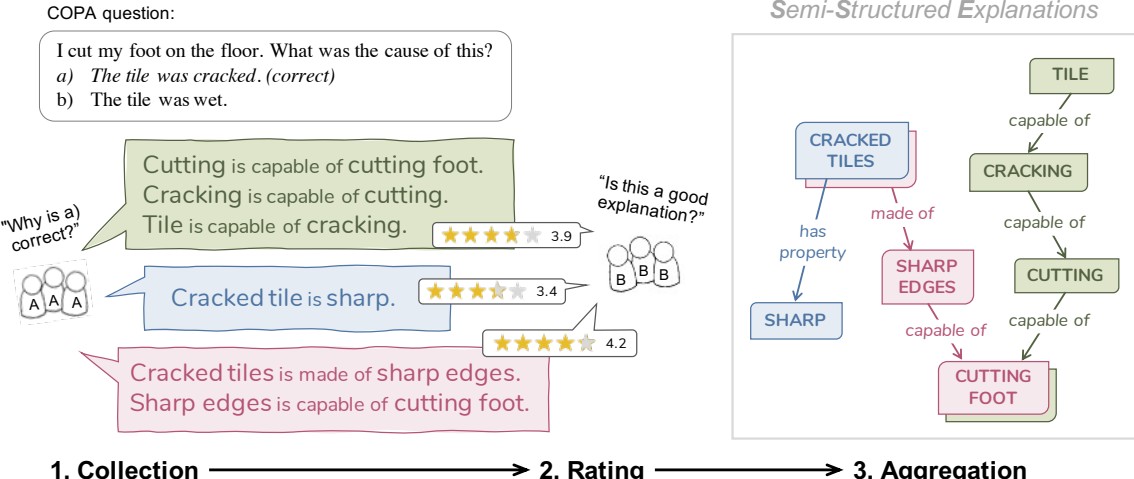

Figure 1: Construction of COPA-SSE. Crowdworkers gave one or more triple-like statements explaining the correct answer (1) which were then rated by different workers (2). Each statement consists of head and tail text linked by a ConceptNet relation. The statements can be aggregated into an explanation graph (3).

Specifically, we introduce Semi-Structured Explanations for COPA (COPA-SSE), a new explanation dataset for the Choice of Plausible Alternatives (COPA) dataset [Roemmele et al., 2011].[1] Each explanation consists of a set of English statements, which, in turn, consist of a head text, a selected predicate, and tail text, mimicking ConceptNet [Speer et al., 2017] triples (Figure 1). The head and tail texts are free-form, allowing an open concept inventory. Each explanation also includes quality ratings. COPA-SSE is the starting point of ongoing work on improving graph-based approaches to commonsense reasoning with a focus on improving relevant knowledge extraction. In this paper, we introduce COPA-SSE (§2), detail its construction (§3), and discuss future use cases (§4). COPA-SSE is available at https://github.com/a-brassard/copa-sse.

## 2. Semi-Structured Explanations for COPA

**Design goals.** Our goal is to add high-quality explanations to Balanced COPA. Since the nature of a good explanation is subject of debate [Miller, 2019], we adopt a working definition: A good explanation is a minimal set of relevant common sense statements that coherently connect the question and the answer. For example, the fact *Opening credits play before a film.* connects the question *The opening credits finished playing. What happened as a result?* and its answer *The film began.* Commonsense KGs such as ConceptNet provide such statements but have limited coverage [Hwang et al., 2021]. For example, even if question and answer concepts are found in the KG, the paths between them can degenerate into long chains of statements that are neither minimal nor relevant (Figure 2). In contrast to structured approaches, unstructured free-form text is not limited by KG coverage. Previ-

---

1. We used the Balanced COPA [Kavumba et al., 2019] variant whose modifications counter superficial cues. Its questions are a superset of the original COPA dataset.

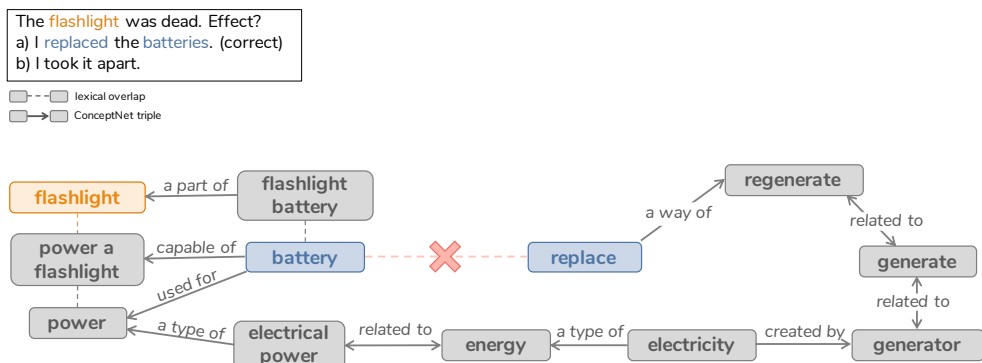

Figure 2: Illustrative example of a structured explanation, manually extracted from ConceptNet. One author attempted to find paths connecting question and answer concepts, but was unable to find a meaningful path between *battery* and *replace.* The two concepts are connected, but the path contains irrelevant facts to the point of being meaningless.

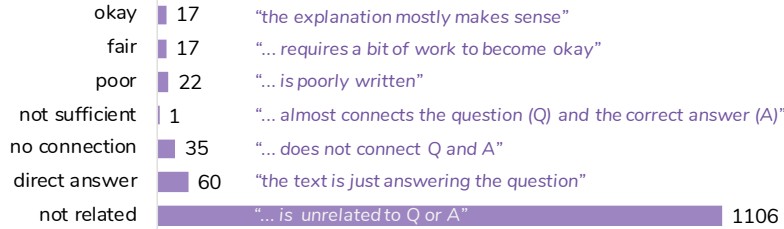

Figure 3: Results of a manual inspection of 1200 CoS-Eexplanations.

ous work has elicited such free-form explanations from crowdworkers, but suffers from low quality. For example, in a manual inspection of 1,200 CoS-E samples most explanations were judged to be unrelated and only a small fraction were deemed acceptable explanations (Figure 3). Aiming for a golden middle, we devise a semi-structured explanation scheme comprising a set of triple-like statements. Each statement consists of open-ended head text and tail text connected with a ConceptNet relation. In practice, crowdworkers created explanations by selecting a predicate from a list while providing free text for the two concept slots.[2] This format encouraged workers to provide explanations close to our definition without being restricted to a pre-defined inventory of concepts. We refer to this combination of free text and ConceptNet predicates as *semi-structured explanations*.

**Dataset statistics.** Table 1 shows examples of COPA-SSE explanations. COPA-SSE contains $9,747$ commonsense explanations for $1,500$ Balanced COPA questions. Each question has up to nine explanations given by different crowdworkers. We provide the triple-format described above, as well as a natural language version obtained by replacing ConceptNet relations with the relation texts shown in Table 2. 61% of explanations are only one state-

---

2. While crowdworkers were encouraged to write short texts corresponding to a single idea, some produced longer texts that can be broken down into multiple statements. We discuss this in Section 4.2.

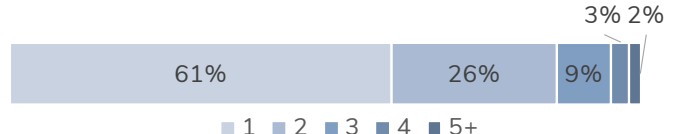

Figure 4: Number of statements per explanation.

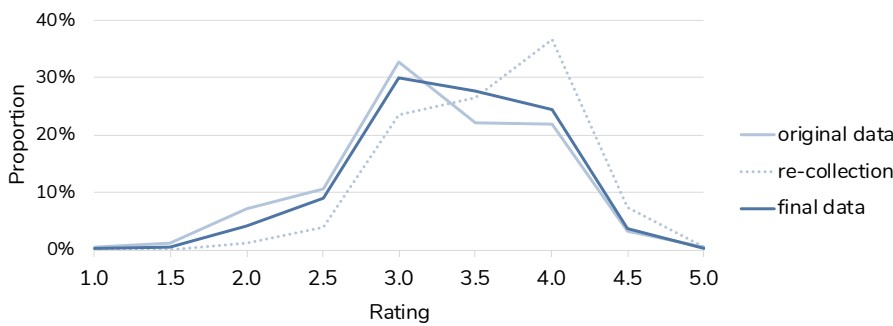

Figure 5: Average rating distribution before (original data) and after the re-collection round (final data). Values are rounded to the nearest half-star.

ment while the other 39% comprise two or more, with the longest explanation being ten statements (Figure 4). Each explanation has a quality rating on a scale of 1 to 5 as given by crowdworkers. Figure 5 shows the rating distribution after initial collection (original data). To guarantee that each Balanced COPA instance is explained by high-quality explanations, we collected additional explanations until most Balanced COPA instances (98%) had at least one explanation rated 3.5 or higher (final data). Initially, 38% of explanation were over this threshold, which increased to 44% after the additional collection run. We kept the lower-quality explanations as they can be useful negative samples. Finally, we aggregated crowdworker explanations by heuristically breaking down long text into shorter statements and by merging similar terms. We now describe each crowdsourcing step in more detail.

## 3. Crowdsourcing Protocol

Crowdworkers were asked to provide one or more statements that connect the question and the answer in a triple format: a free-form head text, a selection of ConceptNet relations, and a free-form tail text, together forming a commonsense statement (§3.1). Each set of statements was then rated by five different workers (§3.2). To gather additional high-quality explanations, we invited workers whose explanations were highly rated to provide additional explanations (§3.3). Section 3.4 lists worker qualifications and compensation.

### 3.1 Collecting Explanations

Figure 6 shows our collection form. Workers were given a COPA question and two answer choices with the correct one marked. The input row below consists of two text fields for inputting concepts and a drop-down box for selecting the relation between them. Workers

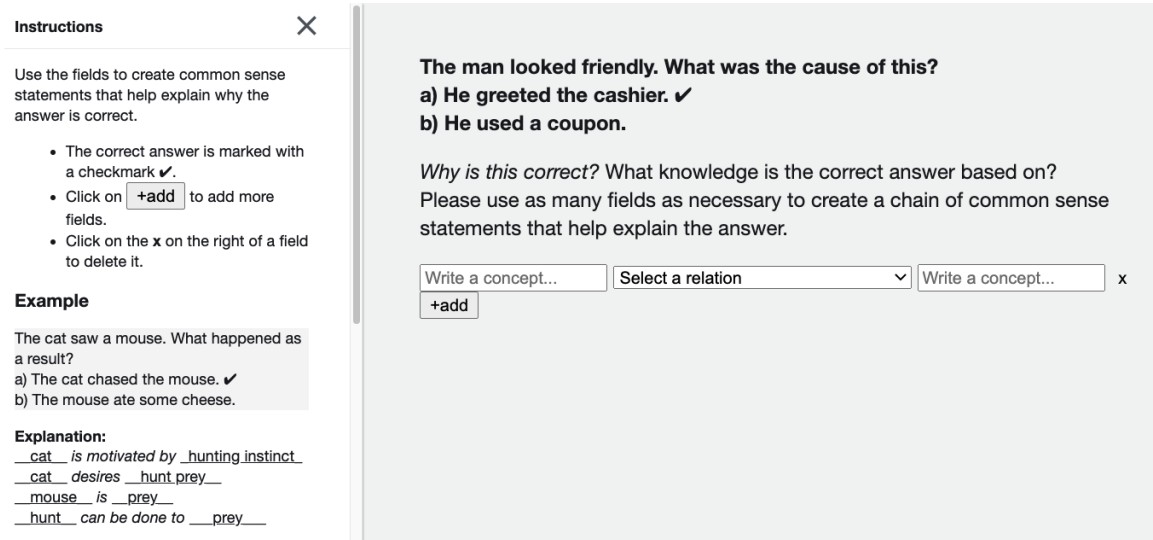

Figure 6: Form for collecting explanations.

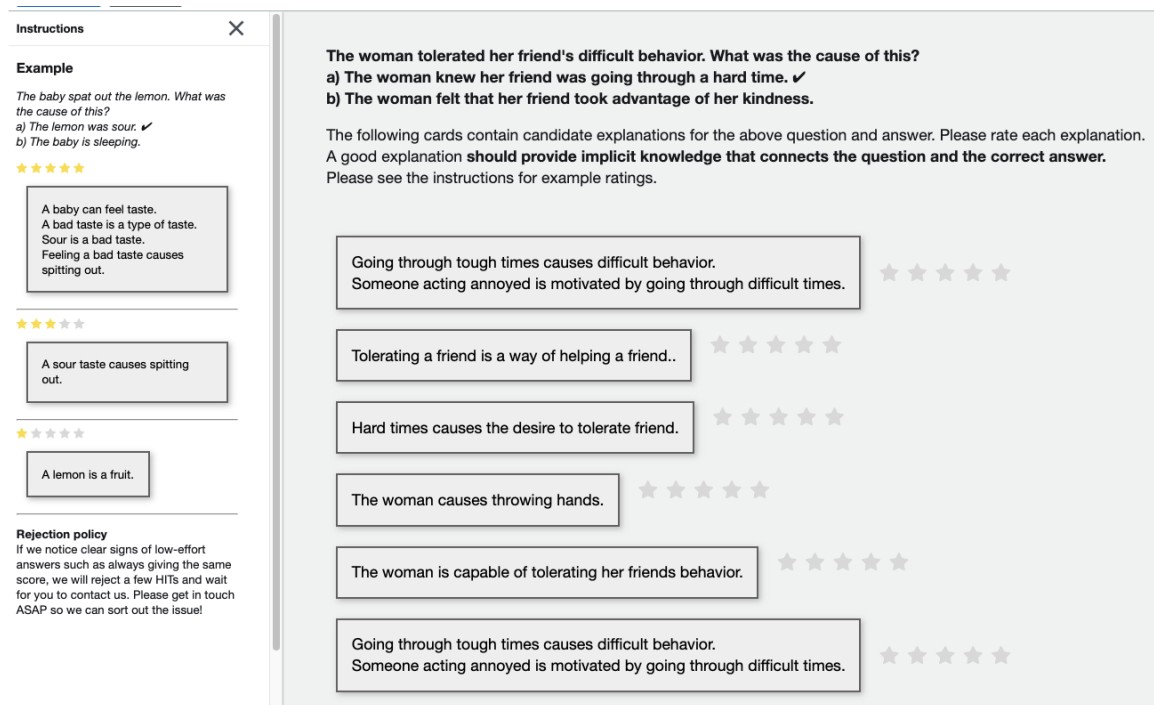

Figure 7: Form for rating Explanation.

| | |
|---|---|
| **The documents were loose. Effect?** | |
| ✓ I paper clipped them together. | ✗ I kept them in a secure place. |

| | |
|---|---|
| ★★★★₁ | Paper clip is used for loose documents. |
| ★★★★ | Paper clips is used for keeping documents together. Paper clipping can be done to have the documents together. |
| ★★★★ | Paper clip is used for clipping paper together. |
| ★★★↲ | Paper clip is used for organizing papers. |
| ★★★↲ | Paper clip can be done to keep papers together. |
| ★★★₁ | The paper clipped is a way of holding the papers together. |

| | |
|---|---|
| **The girl met her favorite actor. Effect?** | |
| ✓ She asked him for his autograph. | ✗ She went to see his new film. |

| | |
|---|---|
| ★★★★ | Favorite actor causes the desire to get autograph. |
| ★★★★ | Asking for an autograph is a part of meeting your favourite actor. Meeting your favourite actor causes the desire to ask for an autograph. |
| ★★★★ | Girl has a fantasy. Favorite actor causes fantasy. Autograph is motivated by fantasy. Girl is capable of asking for. Girl is located near her favorite actor. |
| ★★★★ | Meeting her favorite actor causes the desire to ask for an autograph. |
| ★★★↲ | Asking for autograph is a way of meeting favorite actor. |
| ★★★₁ | Autographs is used for meeting famous people. |
| ★★★ | Actors causes the desire to obtain autographs. |
| ★★★ | A girl meeting their favorite actor desires to get an autograph. Getting an autograph is motivated by meeting someone famous. |
| ★★★ | Seeing favorite actor causes the desire to want autograph. |

| | |
|---|---|
| **They lost the game. Cause?** | |
| ✗ Their coach pumped them up. | ✓ Their best player was injured. |

| | |
|---|---|
| ★★★★₁ | Game is a team work. Player is a part of a team. Player injured causes team not working properly. Team not working properly causes lose the game. |
| ★★★★ | Best player is a part of the team. Injury of the best player causes the team to lose. |
| ★★★★ | Their best player being injured causes the team to lose. |
| ★★★★ | Teams is made of players. Injuries is capable of causing losses. |
| ★★★★ | Injury is capable of causing loss. |
| ★★₁ | The team causes the injury. |

Table 1: Examples of collected and rated explanations for Balanced COPA questions.

could increase the number of rows to provide explanations with multiple statements, as they were encouraged (but not forced) to do. The relations are a subset of ConceptNet predicates which we selected and translated into English for easier understanding by non-experts (Table 2). For example, the input  _an apple_  is a  _fruit_  corresponds to the statement *"An apple is a fruit."* and the triple (*"an apple"*, IsA, *"fruit"*).

Free-form text guarantees neither consistent granularity nor chains of statements connected by matching concepts. For example, a phrase such as *"the act of eating a sweet fruit"* can be given as tail text, even though the next statement might not include that same phrase. We opted to leave this freedom as longer statements can still form coherent

| Relation text | ConceptNet equivalent | Relation text | ConceptNet equivalent |
|---|---|---|---|
| *– Properties –* | | *– Actions –* | |
| is (has property) | /r/HasProperty | is used for | /r/UsedFor |
| is a | /r/IsA | is capable of | /r/CapableOf |
| is a type of | /r/IsA | is a way of | /r/MannerOf |
| is a part of | /r/PartOf | is created by | /r/CreatedBy |
| has a | /r/HasA | can be done to | /r/ReceivesAction (*) |
| is made of | /r/MadeOf | desires | /r/Desires |
| is located at | /r/LocatedAt | is motivated by | /r/MotivatedByGoal |
| is located near | /r/LocatedNear | | |
| *– Events –* | | *– Word meaning –* | |
| causes | /r/Causes | is related to | /r/RelatedTo |
| causes the desire to | /r/CausesDesire | is similar to | /r/SimilarTo |
| is obstructed by | /r/ObstructedBy | is a symbol of | /r/SymbolOf |
| happens during | /r/HasSubevent (*) | is a synonym of | /r/SynonymOf |
| is the first thing that happens during | /r/HasFirstSubevent (*) | | |
| is the last thing that happens during | /r/HasLastSubevent (*) | | |
| must happen before | /r/HasPrerequisite (*) | | |

Table 2: List of given relations and their ConceptNet equivalent. Relations marked with (*) have their left and right concepts switched when converting to triples, e.g.: *"A can be done to B"* becomes (*"B"*, `ReceivesAction`, *"A"*).

explanations, and, as we found in preliminary runs, introducing strict constraints might lead to unnatural and/or less informative explanations. Overly long statements were rare, as most workers followed the simple examples we provided.

## 3.2 Rating Explanations

Figure 7 shows our form for rating explanations. Each explanation was rated by five workers. Workers were shown a COPA instance and five explanations to rate with up to five stars. As a control, workers had to rate the first explanation again at the end of the HIT, totaling six ratings per HIT. We disregarded (but did not reject) ratings by workers who had more than a one-star difference in this control.[3] Workers were instructed to give a higher rating to explanations containing relevant and more detailed statements and low ratings to uninformative or nonsensical explanations. We observed that detailed, related statements were also low-rated if they did not explain why the answer is correct. Examples of high-rated and low-rated explanations are shown in Table 3. While these ratings serve as generic estimate of quality, we recommend against using them as measurements of any single characteristic such as relevance or thoroughness since they were not defined as such.

## 3.3 Re-collection

To increase the number of higher-rated explanations, we invited workers who provided high-quality explanations to provide additional explanations for a higher fee. We collected four new explanations for questions that had all five explanations rated below 3.5-stars, two new explanations if one was above this threshold, and one new explanation if two were above this threshold. New explanations were then rated in the same way as the original ones.

---

3. We allowed a 1-star difference as one could change their opinion on the first seen explanation after seeing other examples. In case of such a difference, we only retain the last rating.

| | |
|---|---|
| The woman sensed a pleasant smell. Effect? ✓ She was reminded of her childhood. | |
| ★★★★★ Pleasant smell is a way of bring happiness. Happiness causes nostalgia. Nostalgia is related to a smell. Smell causes her to think her childhood. | |
| The flashlight was dead. Effect? ✓ I replaced the batteries. | |
| ★★★★★ Batteries is used for flashlights. Power is created by batteries. Replacing batteries is a way of restoring power. | |
| The car looked filthy. Effect? ✓ The owner took it to the car wash. | |
| ★★★★★ The owner desires clean car. Car wash is used for washing cars. | |
| I got hooked to my conversation with the woman. Cause? ✓ The woman was telling a funny story. | |
| ★★★★★ Humor is capable of creating interest. A funny story is a way of engaging someone in conversation. | |
| The man's pocket jingled as he walked. Cause? ✓ His pocket was filled with coins. | |
| ★★★★★ Jingling whilst walking is created by having a pocket full of coins. Having a pocket full of coins is capable of making a jingling sound whilst walking. | |
| My favorite song came on the radio. Effect? ✓ I sang along to it. | |
| ★ This is a symbol of simple. | |
| The window was opaque. Cause? ✓ The glass was stained. | |
| ★ Many is located at is. | |
| The rain subsided. Effect? ✓ I went for a walk. | |
| ★ The rain has a fresh smell. | |
| The woman tolerated her friend's difficult behavior. Cause? ✓ The woman knew her friend was going through a hard time. | |
| ★₁ The woman causes throwing hands. | |
| The girl was not lonely anymore. Cause? ✓ She made a new friend. | |
| ★₁ Making is motivated by loneliness. | |

Table 3: Top five and bottom five explanations. Highly rated explanations tend to be detailed and explicitly connect the question and answer. Low rated ones are incoherent, completely irrelevant, or related facts but irrelevant as an explanation.

### 3.4 Compensation and qualifications

Workers received $0.30 per explanation in the first collection round and $0.40 in the re-collection round. In the rating rounds, workers received $0.30 for six ratings (five unique and one control). We restricted all our rounds to workers in GB or the US with a HIT approval rate of 98% or more and 500 or more approved HITs. For re-collection, we invited workers whose explanations averaged more than 3.5 stars over ten or more explanations. The total cost, including Amazon Mechanical Turk fees and excluding trial runs, was $8,651.16.

## 4. Discussion

### 4.1 Alternative approach: extracting then filling in

Following our definition, a more obvious hybrid approach might be to first extract existing data from a KG then fill in what is missing. However, we decided against this approach as workers might be biased towards selecting a relevant relation if they see it, potentially resulting in non-minimal explanations. Our goal being to capture *natural* human reasoning,

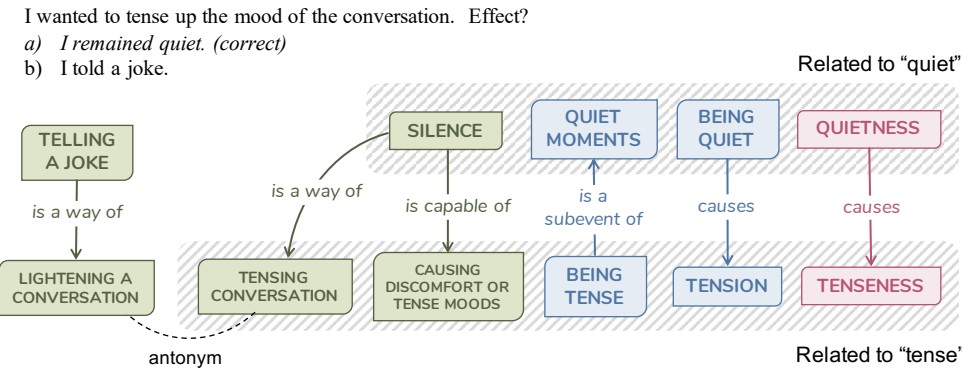

I wanted to tense up the mood of the conversation. Effect?
a) I remained quiet. (correct)
b) I told a joke.

Figure 8: Statements from three different explanations referring to similar or related concepts.

we borrowed the language of existing resources but only as a vehicle for expressing one's reasoning.

## 4.2 Outlook: COPA-SSE as a Resource for Commonsense Reasoners

We created this dataset with several uses in mind: it can serve as training data for (textual) explanation generation models, or as representations of "ideal" subgraphs to use as gold data for graph-based reasoners or to compare with existing KGs. While COPA-SSE already contains ConceptNet-like data, aggregating the explanations into a single, connected graph requires some additional post-processing.

**First steps—aggregating the statements into a unified graph.** An explanation occasionally has complex head or tail texts, and explanations for the same question may refer to the same concepts with different surface forms (Figure 8). One approach to aggregating them is to break down and normalize the concepts into a more uniform granularity. As a preliminary attempt, we extracted the root of a complex concept then heuristically matched the remaining information to relations by breaking down simple noun phrases into objects and their properties, and separating common forms of action descriptions into the agent, location, and object. This nearly halved the number of nodes while the edge count stayed about the same, indicating there was a significant amount of cases where nodes referred to a similar concept (Table 4). Another approach would be to directly ground the explanations, i.e., matching to ConceptNet concepts where possible then adding the rest. We will explore best practices as we advance in our research. The preliminary version of the aggregated graphs are also included with the dataset.

**Final goal—improving commonsense reasoning.** COPA-SSE's textual explanations can be used to improve LM-based systems such as Commonsense Auto-Generated Explanations (CAGE) [Rajani et al., 2019], the system CoS-E was first intended for, which uses a LM to generate explanations as an intermediate step during training and inference. Its triple-like format can in turn be useful for improving graph-based reasoning systems. For example, the current best-performing graph-based system on the CommonsenseQA benchmark [Talmor et al., 2019], QA-GNN [Yasunaga et al., 2021], extracts a relevant subgraph

|  | Before | After |
|---|---|---|
| Node count | 16.7 | 9.6 |
| Edge count | 10.4 | 10.1 |
| Node degree | 1.3 | 2.1 |

Table 4: Node and edge statistics before and after additional post-processing. All values are averages.

from ConceptNet, scores the relevance of each node, uses a graph neural network (GNN) to represent the knowledge, then uses combined information from the GNN and a LM's representation of the question-answer context to predict the answer. The triple-form explanations can be considered as highly relevant subgraphs and can thus be used for supervised training of subgraph extraction and relevance scoring. Even though the explanations are in a similar format, their degree of freedom made it possible to collect new information that might not have been present in ConceptNet. We intend to explore the difference in performance using (i) our explanations as gold graphs, (ii) the commonly used method of extracting topic terms and its n-hop neighbors in KGs [Lin et al., 2019], (iii) a subgraph extraction method additionally trained with our graphs, and (iv) generated graphs, all using the same reasoning module. Comparing these settings will measure the effect of incomplete or unweighted knowledge graphs in graph-based commonsense reasoning and pave the way to more flexible systems.

## 5. Conclusion

We introduced a new crowdsourced dataset of explanations for Balanced COPA in a triple-based format intended for advancing graph-based QA systems and clear comparison with existing commonsense KGs. The dataset provides relevant and minimal information needed to bridge the question and answer. Our dataset includes explanations in text form and raw triple form as written by crowdworkers, and post-processed versions with compound concepts broken down into simpler statements. This dataset is the starting point for ongoing work on improving graph-based QA systems based on extracting subgraphs from KGs, scoring their relevance, then reasoning with a graph neural network.

## Acknowledgments

This work was partially supported by JST CREST Grant Number JPMJCR20D2.

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
