# OpenReview forum: "COPA-SSE: Semi-Structured Explanations for Commonsense Reasoning"
_AKBC.ws/2021/Workshop/CSKB — CSKB_

### Official Review · Reviewer_kcuW · 2021-09-15
**Nice dataset, but lacks an extrinsic evaluation**

**Rating:** 6
**Confidence:** 4

**Review:**

This paper describes an explanation dataset crowdsourced through open-ended, template constrained explanations. The hypothesis for template based constraining is well-motivated because templates have been shown to assist in quality control during crowdsourcing [1].

If the ultimate goal is downstream task improvement, then one would really hope for an initial experiment that uses past methods with your new explanation data. This is useful to evaluate the usefulness of copa-sse and more importantly compare it with CoS-E. It would answer two  interesting questions (1) whether despite the noisy open ended data is still sufficient to train a commonsense reasoning model; and (2) does imposing structure add value to the extrinsic task.

Clearly, adding structure would be more human readable. But from a modeling perspective is it necessary to aggregate the explanations into a unified graph? Some edges in the graph may be less reliable than others and a model can learn these signals through training (you have nice negative examples). If you were to aggregate into a graph, can the method remove an unreliable edge faithfully e.g., can it lead to a globally inconsistent structure?

It was really surprising to note that more than 90 percent of entries in CoS-E are irrelevant to the question or the answer. I checked the CoS-E paper which acknowledged that it was difficult to control the quality of open-ended annotations, but they performed some quality checks. Besides, they found that 58 percent of the entries contain the ground truth (so it is surprising that manual inspection found them to be completely unrelated to the answer).

The presented dataset is of a decent size (1500 questions) with 70 percent examples having a human agreement. Ideally, the dataset statistics/size should be based on the number of questions rather than the number of explanation statements. e.g., how many questions are in the subset with a majority agreement? Until we have any basic extrinsic evaluation numbers, it is hard to tell whether this dataset size is sufficient.

Typos etc.: The paper is generally well-written, there are some typos such as: "for an higher fee (section 3.3)"

References:

[1] A Dataset for Tracking Entities in Open Domain Procedural Text. Tandon et al EMNLP 2020 (https://aclanthology.org/2020.emnlp-main.520.pdf)

---

### Decision · Program_Chairs · 2021-09-18

Accept